# Protocol for DEprescribing and Care to reduce Antipsychotics in Dementia (DECADE)–A hybrid effectiveness-implementation pilot study

Kirstine Skov Benthien[1,2]*, Laura Victoria Jedig Lech[3], Hanne Birke[2], Sidsel Maria Jørgensen[2], Tina Andersen[3], Stine Vest Hansen[3], Jon Trærup Andersen[4,5], Kristian Karstoft[4,5], Michaela Schiøtz[2], Charlotte Vermehren[4,6,7]

1 Palliative Care Unit, Copenhagen University Hospital–Hvidovre, Hvidovre, Denmark, 2 Center for Clinical Research and Prevention, Copenhagen University Hospital–Frederiksberg, Frederiksberg, Denmark, 3 Hillerød Municipality, Denmark, 4 Department of Clinical Pharmacology, Copenhagen University Hospital–Bispebjerg, Copenhagen, Denmark, 5 Department of Clinical Medicine, Faculty of Health and Medical Sciences, University of Copenhagen, Copenhagen, Denmark, 6 Department of Drug Design and Pharmacology, Faculty of Health and Medical Sciences, University of Copenhagen, Copenhagen, Denmark, 7 Hospital Pharmacy, Capital Region, Herlev, Denmark

* kirstine.skov.benthien@regionh.dk

**Data Availability Statement:** No datasets have been generated or analysed yet.

## Abstract

### Introduction

Behavioural and psychological symptoms of dementia (BPSD) should only rarely and briefly be treated with antipsychotics. Despite recommendations to the contrary, the use of antipsychotics in nursing home residents with dementia is widespread and followed by serious adverse effects. Intervention studies on methods to reduce the use of antipsychotics in persons with dementia are few and needed. The aim of this protocol is to describe the rationale and content of the intervention DEprescribing and Care to reduce Antipsychotics in DEmentia (DECADE)–a hybrid effectiveness-implementation pilot study.

### Materials and methods

This is a protocol of a prospective hybrid effectiveness-implementation pilot study. The primary aim of DECADE is to reduce the use of antipsychotic drugs by 50% in 50% of nursing home residents with dementia while maintaining or improving BPSD. The intervention is implemented in six nursing homes including approximately 190 residents with dementia and consists of Academic Detailing, medication review, education of nursing home staff, and care plans. The evaluation of feasibility and potential effectiveness is an overall assessment of all clinical and process outcomes. Logistic regression analyses will be used to investigate factors characterizing situations with prescription of antipsychotics. BPSD is analysed with a before- and after design using self-controlled case series methods and the use of antipsychotics is analysed as interrupted time series.

**Funding:** The interventions are funded by the Danish Board of Health through a grant received by CV. The funders had and will not have a role in study design, data collection and analysis, decision to publish, or preparation of the manuscript.

**Competing interests:** The authors have declared that no competing interests exist.

## Discussion

This protocol describes a study that will provide an indication of DECADE effectiveness and a model for upscaling and further evaluation in a controlled design.

## Introduction

Almost 90% of persons diagnosed with dementia will experience Behavioural and Psychological Symptoms of Dementia [BPSD] [1–3] and some develop severe symptoms such as psychosis, aggression, agitation, and depression. The symptoms can be triggered by lack of sleep, pain, malnutrition, or changes in the environment [4]. Recommendations for the management of BPSD therefore focus on non-pharmacological interventions such as psychological, psychosocial end environmental interventions [5, 6]. Non-pharmacological interventions include developing and regularly re-evaluating person-centred treatment goals, which is important in persons with dementia [5, 6]. For persons experiencing severe BPSD such as extroverted physical or verbal aggression, antipsychotic drugs may be used for shorter or longer periods with a significant, but small effect on symptom severity and caregiver load [7]. It is estimated that over 20% of all persons with dementia receive antipsychotic drugs to treat BPSD, not including antipsychotic drugs for other mental illnesses [8]. In nursing homes, the use of antipsychotic drugs in persons with dementia is even higher with up to 40% of all residents receiving antipsychotics [8]. Furthermore, almost 50% of persons with dementia treated with antipsychotics are treated for more than a year, despite the lacking evidence of clinically relevant effect for long term use [9–11]. Despite the indication for use, many persons with dementia treated with antipsychotics experience little or no effect of the treatment on BPSD [12]. Further, almost all persons experience side-effects, of which some are severe including falls, extrapyramidal symptoms, and cardiovascular adverse events [13]. Observational studies have found an increased mortality of up to 30–35% in persons with dementia who are treated with antipsychotic drugs [14–16]. In summary, the need to reduce an irrational use of antipsychotic drugs in persons with dementia is urgent. The HALT study tested deprescribing of antipsychotics alongside providing nursing home staff with knowledge on non-pharmacological interventions through education and general practitioners with best practice guidelines and recommendations through Academic Detailing [AD] [17]. The intervention was successful in initially deprescribing antipsychotics, but the drugs were too often re-prescribed due to requests from nursing home care staff [18]. The COSMOS cluster-RCT evaluated a multi-component nursing home intervention and Spek et al. evaluated biannual medication reviews in an RCT and both trials demonstrated significant reductions in other psychotropic drugs but not antipsychotics [19, 20]. Overall, intervention studies on methods to reduce use of antipsychotics in persons with dementia are few and called for. In consequence, the aim of this protocol is to describe the rationale and content of the intervention DEprescribing and Care to reduce Antipsychotics in DEmentia (DECADE)–a hybrid effectiveness-implementation pilot study.

## Materials and methods

### Design and outcomes

This article describes the protocol for the one-year prospective DECADE pilot study before data regarding residents with dementia are collected. Thus, there are no data available based on this manuscript.

The primary aim of DECADE is to reduce the use of antipsychotic drugs by 50% (as a 50% reduction in Defined Daily Dose (DDD)) in 50% of nursing home residents with dementia while maintaining or reducing BPSD. The choice of a 50% reduction in 50% of nursing home residents instead of 100% / 100% is based on previous experiences and reflects the realization that, in some cases with severe BPSD or pre-existing psychiatric disorders, deprescription is not in the best interest of the nursing home resident. The specific combination of intervention components in this study has not been evaluated in any trial, but they reflect evidence-based guidance [4]. Despite the risk in antipsychotics and non-pharmacological interventions being considered first choice in BPSD-care, implementation remains a challenge. Parallel to effect measurements, implementation will be evaluated with an assessment of fidelity and acceptability, including measurements of Academic Detailing, medication review, education of staff, and care plans. Through this dual focus on effect- and implementation outcomes, we classify this study as a hybrid effectiveness-implementation pilot study, type two, as described by Bauer et al. [21].

## Setting

The setting is nursing homes in Hillerød Municipality in Denmark–a municipality with approximately 53.000 inhabitants in the Northern part of Zealand. The Danish healthcare system is universal and tax financed. Primary care consists of general practitioners (GPs) that act as gate keepers to hospital-based healthcare, and of nursing homes and home nursing which are provided by 98 municipalities. Hillerød municipality manages six nursing homes included in this study with varying sizes, staff skill mix (registered nurses, nurses' aides, and allied health professionals), and resident composition and 17 general practices staffed by a total of 22 GPs.

## Participants

Participant inclusion criteria are dementia diagnosis or investigation of dementia and nursing home residence. The number of participants is limited to the residents of the six nursing homes in Hillerød municipality, wherein all residents are included. In these nursing homes, approximately 190 residents have a diagnosis of dementia (D-Group). A subsection of approximately 60 persons with dementia who use antipsychotic medication (DA-Group) at the time of project start. All nursing home staff (S-Group) with direct resident contact or supervisory capacity are eligible to participate in a staff survey and the intervention. A total of 22 GPs has the responsibility of the medical treatment in the six nursing homes.

## Intervention

The intervention consists of four components aimed at GPs and nursing home staff presented in the following (Academic Detailing, Education, Medication Review, and Care Plans). The intervention was developed in an iterative process including the research team and staff from Hillerød Municipality (see Acknowledgments) and agreed on in the steering group of the project.

## Academic detailing

GPs in Hillerød, who are responsible for the medical treatment of nursing home residents, are educated in rational antipsychotic prescribing and deprescribing of irrational use of antipsychotics through the method of Academic Detailing. Academic Detailing is an evidence-based method aiming to improve the quality of prescribing in general and thereby improve treatment to positively affect health outcomes [22]. In this project, Academic Detailing is provided

by a pharmacist. The pharmacist carries out the detailing in a 15 minutes face-to-face visit with the GPs at their clinics [23]. The dissemination focuses on the evidence-based recommendation of deprescribing of antipsychotics and the use of non-pharmacological actions aiming at reducing the use of antipsychotics. The pharmacist uses education material developed by the research team, that provides a simple overview of the evidence-based guidelines on prescribing and deprescribing of antipsychotic drugs in persons with dementia.

## Education

The study includes the development and implementation of two sessions of education for nursing home staff: general education and adapted education. The purpose of the first general education session at the beginning of the project is to increase the knowledge about the rational use of antipsychotics and non-pharmacological interventions and to be professionally prepared to reduce the use of antipsychotic drugs among the included residents. The session will include updating on evidence-based practice regarding care of persons with dementia, effects and adverse events of antipsychotics, reasons to cease or reduce use, behavioral and psychological symptoms of dementia, management of BPSD, and person-centered care inspired by the principles of Tom Kitwood and Marte Meo [24, 25]. The session lasts 3–4 hours, is planned by the research team and municipality, and held by a multidisciplinary team consisting of a geronto-psychiatrist, a GP consultant, a clinical pharmacist at the municipality, a nurse specialized in dementia care, and an occupational therapist including the research team. To ensure participation of all staff, the session will be repeated four times. To fit the education to the staff's needs, the content of the subsequent adapted session will be informed by knowledge from qualitative analyses, baseline survey results and the medication reviews.

## Medication review in network groups

Medication reviews are carried out for each resident to ensure a rational medication use. This involves a systematic and critical review of the nursing home resident's drug therapy, with a special focus on reducing the use of antipsychotics and at the same time ensuring that this does not lead to an increased prescription of other psychotropic drugs such as benzodiazepines and antidepressants. The medication reviews are conducted as an educational-oriented intervention performed by multidisciplinary network groups that consist of the resident's GP, nursing home staff close to the resident in conjunction with the medication team consisting of a geronto-psychiatrist, a GP consultant, and a clinical pharmacist from the Department of Clinical Pharmacology. The medication team prepares the medication reviews before they meet with the rest of the network group to discuss and plan any medication changes. The patient's drug therapy is reviewed using the resident's up-to-date medication list, patient records with information on BPSD, diagnoses, contraindications, and potential drug-drug interactions. The resident's GP makes the final decision regarding the changes to the medicine in line with the general role of GPs agreed by the GPs union and the Regions who oversee the healthcare system. Following the medication review network meetings, the resident's GP discusses the medication changes with the resident or informal caregiver as per standard practice.

The 7-step procedure and method of medication review in network groups are described in depth by Frandsen et al. [26]. However, for the present study, some changes will be made to the original procedure and the interprofessional team due to the nature of the residents involved. Unfortunately, it is rarely considered possible to discuss any medication changes with the residents due to their cognitive impairment. Instead, the resident's contact person at the nursing home will be involved in the discussions about the suggested medication changes. The accepted changes to the drug therapy will be registered in the care plans and the degrees

of implementation of medication changes accepted by the resident's GP will be followed up after 3, 6 and 9 months.

### Care plans

A person-centred care plan will be developed for each resident with dementia and current use of antipsychotics. The care plans include assessment of individual causes of BPSD and preventive measures, decisions from medication review, plan for prescribing and reduction of antipsychotics, and strategies for non-pharmacological measures. The care plan is developed by the nursing home residents' regular care team supported by local key personnel and municipal consultants with expertise in dementia to support integration of the theoretical knowledge from the educational sessions into clinical practice. Informal caregivers are offered to be involved in the care plan whenever possible and the care plans are documented in the electronic patient record. The key personnel will also support adherence and follow-up through peer-to-peer supervision and regular conferences. The care plans may be continuously adapted to the needs of the nursing home residents.

### Implementation process

To support an evidence-based implementation strategy, the implementation of the study intervention will be structured with inspiration from The Quality Implementation Framework [QIF] by Meyers et al. [27]. The activities are outlined in Fig 1 presenting the four phases as suggested by the QIF [27].

### Phase 1. Assessment and project organization

To tailor the intervention to fit the context of the nursing homes in Hillerød municipality, local practices are explored through qualitative interviews with selected healthcare staff

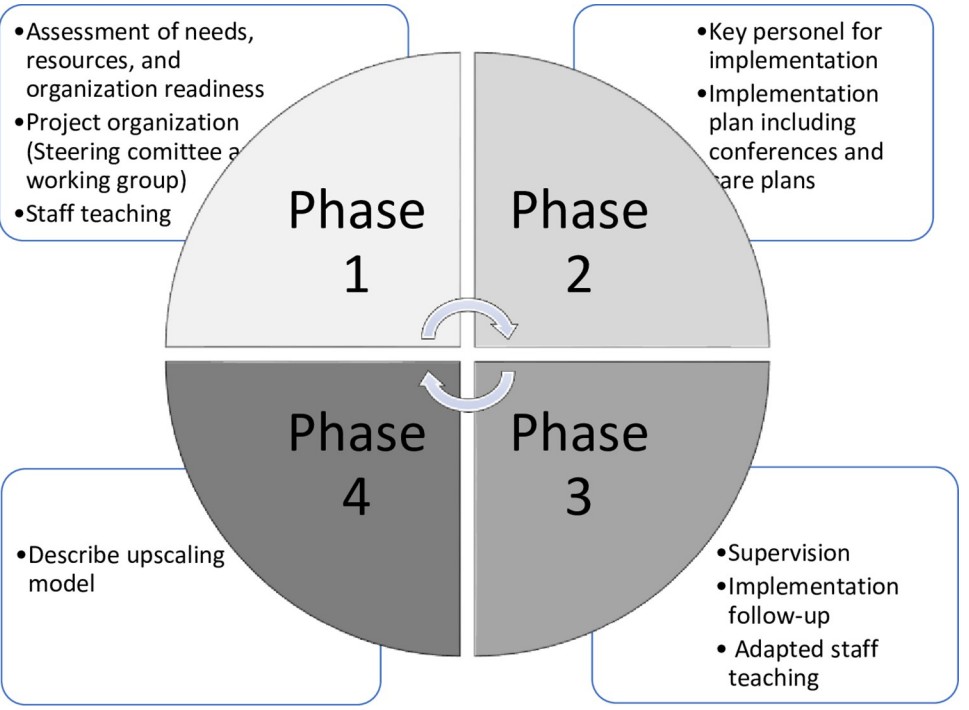

**Fig 1. Quality implementation framework phases and activities.**

employed by nursing homes or rehabilitation centre. To secure a comprehensive perspective of the workflow processes the selection was based on purposive sampling including healthcare workers, nurses, leaders, and key management personnel. The interviews focused on decisions for prescribing and administering antipsychotics and intervention fit (workflow process and resources). The study includes a questionnaire survey with healthcare staff employed by the nursing homes to explore knowledge and self-efficacy. Conditions for prescription of antipsychotics will be explored by reviewing patient records. The project is organized with a steering group representing all sectors including management from Hillerød municipality, a general practice coordinator, the research team from the Department of Clinical Pharmacology and the Center for Clinical Research and Prevention, Bispebjerg and Frederiksberg Hospital, and a representative from the patient organization 'The DaneAge Association' ('Ældre Sagen'). The steering group is responsible for supporting data availability, appoints informants, supervises supportive staff in a working group and secures implementation sustainability.

## Phase 2. Implementation: Key personnel, conferences, and care plans

Key personnel appointed at each nursing home is tasked with securing the development of care plans. The key personnel are supported by municipal consultants with expertise in dementia. Implementation will be secured through nursing home residents' conferences, peer-to-peer support, and follow-up on the tasks defined in the care plans.

## Phase 3. Implementation follow-up and adaptation

The knowledge derived from phase 1 will be used to adapt education activities and possibly identify other new activities to support deprescribing, maintaining or improving BPSD. Furthermore, any implementation shortcomings identified during phase 2 will be addressed to ensure full implementation.

## Phase 4. Upscaling model

The degree of implementation in the six nursing homes will be compared, and a model drawing on the activities from the nursing homes that most convincingly managed to reduce antipsychotics and maintain or improve BPSD will be used to describe a model for upscaling in other nursing homes. That includes conditions perceived as necessary for successful implementation.

## Data

An overview of quantitative data and timing of assessments is presented in Fig 2 inspired by the SPIRIT guideline [28].

## Use of antipsychotics and other psychotropic drugs

Use of antipsychotics and other psychotropic drugs in all nursing home residents with dementia in Hillerød from 2020 to 2023 (D-group) is extracted from registers. This includes drugs, dose, and regular prescriptions. From the DA-group, the circumstances surrounding prescriptions of antipsychotics will be analysed, and the following characteristics will be included: setting (hospital/rehabilitation/nursing home/private home before nursing home), type of care staff involved (nurses/nurses assistants/nurses aids), care staff affiliation (permanent employee/temp), type of physician (GP/hospital doctor/other), presence of nursing home GP, time of day (day/evening/night), and time since diagnosis of dementia. Use of antipsychotic and other psychotropic drugs are available from medical charts.

| Assessment | -T2 | -T1 | | T0 | T1 | T2 | T3 |
|---|---|---|---|---|---|---|---|
| **Activity/ assessment** | **2 years** **follow-back** | **1 year** **follow-back** | **Pre-study** **Screening** | **Baseline** | **3 months** **Follow-up** | **6 months** **Follow-up** | **9 months** **Follow-up** |
| **All residents with dementia in Hillerød nursing homes (D-Group)** | | | | | | | |
| **Antipsychotic drugs** | x | X | | x | | | x |
| **Other psychotropic drugs** | x | x | | x | | | x |
| **Hospital admissions** | x | X | | x | | | x |
| **Survival** | | | | | | | x |
| **Assessments and activities for residents with dementia with current use of antipsychotics (DA-Group)** | | | | | | | |
| **Screening log** | | | x | | | | |
| **Socio-demographics** | | | x | | | | |
| **BPSD** | | | | x | | | x |
| **Medication review** | | | | x | x | x | x |
| **Care plans** | | | | x | x | x | x |
| **Antipsychotic drugs** | | | | x | x | x | x |
| **Antipsychotic drugs as needed** | | | | x | x | x | x |
| **Other psychotropic drugs** | | | | x | x | x | x |
| **Staff (S-Group)** | | | | | | | |
| **Consent** | | | x | | | | |
| **Screening log** | | | x | | | | |
| **Demographics** | | | | x | | | x |
| **Knowledge and competences** | | | | x | | | x |

**Fig 2. Assessment timeline.**

## Assessment of BPSD

Assessment of BPSD will be based on The Neuropsychiatric Inventory (NPI) that includes 12 symptoms, severity, and impact on staff. The Danish version of NPI has demonstrated satisfactory to perfect validity and reliability [29]. Due to time constraints in the nursing homes and to secure a sustainable clinical use of BPSD screening, we chose an abbreviated version—the NPI-Q (NPI-Questionnaire), which is cross-validated with the standard NPI [30]. In this project, BPSD is assessed by the staff in nursing homes with knowledge about each resident in resident conferences.

## Assessment of knowledge and competences by nursing home staff

To examine the impact of the intervention on the self-rated knowledge and competences of the healthcare professionals a repeated measurement with questionnaires is conducted. At baseline, a paper version of the questionnaire is completed by nursing home staff before the

general education day. A follow-up of the survey is conducted at nine months. The survey includes 56 items in total including demographics and three components of the Danish questionnaire Dementia Friendly Hospital Education Assessment Questionnaire (DeFHEAQ), which are used to assess self-rated knowledge (16 items), attitude (11 items), and competence (11 items) of the professional caregivers at the nursing homes [31]. With permission from the author, the word 'hospital' was replaced with 'nursing home'. Finally, the survey includes 12 intervention-specific questions related to the caregiver's knowledge and competences regarding BPSD, antipsychotic treatment and three non-pharmacological methods. The intervention-specific questions are developed by project staff and have not undergone psychometric evaluation.

## Covariates

The following covariates are extracted from the electronic patient record: Age, gender, time since diagnosis of dementia, and comorbidities.

## Analyses

Since the study aims to explore the potential for effectiveness and upscaling of the intervention, no power calculation has been performed. The hypothesis of potential effectiveness will be generated through an overall assessment of all clinical and process outcomes, emphasizing a reduction in use of antipsychotics while maintaining or improving BPSD. The analyses include three areas described below.

## Organisation

Interviews with healthcare professionals during implementation phase 1 will be audio-recorded and transcribed verbatim. Transcribed interviews will be analyzed using systematic text condensation [32]. In the S-Group, healthcare professional knowledge and competences are surveyed and described before and after the intervention with descriptive statistics.

## Antipsychotic prescription practice over time

This study includes all persons with dementia residing in nursing homes in Hillerød between 2020 and 2023 (Group D). Logistic regression analyses will be used to investigate factors characterizing situations with prescription of antipsychotics. Since data on use of antipsychotics, hospital admissions, and survival are systematically registered administrative data, they are available before implementation initiation. Therefore, the historic development in this area may be accounted for and these data are analysed as interrupted time series [33].

## Effectiveness and implementation

This study includes approximately 60 persons with dementia and use of antipsychotics during the project period (Group DA). The potential for intervention effectiveness is assessed with BPSD with a before- and after design using self-controlled case series methods [34]. Decisions from medication review and care plans are followed-up at 3, 6, and 9 months by reviewing patient records. If the deprescription decisions are not applied, GPs will be contacted to ensure follow-through, and the lack of implementation may be studied by qualitative interviews.

## Ethics

**Intervention.** Since reduction of antipsychotics and use of non-pharmacological measures in persons with dementia are considered best practice and ideally should be regular

healthcare, the intervention does not need ethical board approval or participant consent (Assessment F-22056825, Nov. 11[th], 2022, The Capital Region Board of Ethics). Reducing the use of antipsychotics involves the potential risk of medication transfer to other psychotropic medications. This risk will be managed with full medication reviews.

**Data handling.** Data handling and review of patient records require consent or administrative approval from the Centre for Regional Development. Due to cognitive impairment, the nursing home residents with dementia could not consent to the review of patient records. Therefore, the Centre for Regional Development has approved the review of patient records (R-22015601). Healthcare professional participants were informed and provided written consent to the secure handling of the data collected during interviews and surveys according to the Data Protection Regulation.

## Discussion

This pilot study will evaluate a complex intervention aiming to reduce antipsychotics while maintaining or improving BPSD in nursing home residents with dementia. The dual focus on deprescribing and BPSD is set to ensure sustainable changes in clinical practice. We hypothesized that deprescribing antipsychotics without non-pharmacological interventions in lieu of drugs would lead to worsening BPSD, prompting nursing home staff to request prescriptions of antipsychotics once again. Based on previous studies [18], even if deprescription is achieved, the effects will require continuous monitoring to be sustained. We therefore focus on the skills of healthcare professionals and embedding the intervention and follow-ups in the organizational structure.

The study is set in a Danish context involving one municipality, six nursing homes, general practitioners, health care professionals, a pharmaceutic hospital department and a research unit. The intervention components are adapted to the local organizational structure and staff education needs. Therefore, the intervention may not be directly transferred and generalized to other settings. Transfer and upscaling could build on adaptation of principles and implementation of function as suggested by the ADAPT guideline [35]. A close collaboration between researchers, specialists, general practitioners, and health care professionals supports an interdisciplinary development of the intervention focusing on the implementation process and evaluation. The qualitative results give valuable information about the nursing home staffs knowledge and beliefs in the care and treatment of nursing home residents with dementia–useful when implementing initiatives in other nursing homes.

The prospective study design enables nursing home residents with dementia to act as their own control group in the evaluation of BPSD, thereby eliminating all time-invariable confounding. This pilot study is explorative and therefore without power calculation. In consequence, focus is not on statistical significance that is subjected to risk of type II-errors but instead on effect sizes and generating a hypothesis of potential effectiveness or lack thereof.

For evaluation purposes, the baseline BPSD assessment of the nursing home residents with dementia should take place before the four intervention components are implemented. However, since the nursing home staff needs an introduction to perform the BPSD assessments, this is included in the education sessions. Through the education sessions, the nursing home staff may be inspired to implement non-pharmacological measures for preventing and managing BPSD in the time frame before the baseline BPSD assessment, which may be improved. The timing of education may therefore lead to an underestimation of the effect size in BPSD. However, since the study aims to improve or maintain BPSD, this bias will not affect the overall judgement of potential efficacy.

The use of antipsychotics has been a continuous focal point for patient organizations and healthcare providers. Without a control group, other initiatives not related to the intervention

may reduce the use of antipsychotics. To account for the historic development in use of anti-psychotics, prescriptions will be included two years before the intervention implementation.

The research group will be involved in all phases and content of the data collection and analysis to ensure appropriate discussions and reflexiveness. The research group will consult the steering group of specialists from clinical hospital departments and universities to support the analysis and dissemination of results. A communication strategy consisting of publication in peer-reviewed scientific journals, shorts reports targeting practitioners, stakeholders, patient organisations, and policymakers, and presentations at conferences and meetings will be followed to provide sharing of key findings and implications for practice.

This study will deliver an indication of DECADE effectiveness and a model for upscaling and further evaluation in a controlled design. The DECADE intervention consists of four components that may interact. Therefore, the effect of each component cannot readily be isolated. The model for upscaling will include the components, that were convincingly implemented and therefore presumably active components in case of effectiveness.

## Acknowledgments

The authors wish to thank Susie Dybing for participating in intervention definition and motoring the implementation and the steering group for overseeing and supporting the project.

## Author Contributions

**Conceptualization:** Kirstine Skov Benthien, Hanne Birke, Jon Trærup Andersen, Kristian Karstoft, Michaela Schiøtz, Charlotte Vermehren.

**Funding acquisition:** Kirstine Skov Benthien, Jon Trærup Andersen, Kristian Karstoft, Michaela Schiøtz, Charlotte Vermehren.

**Investigation:** Charlotte Vermehren.

**Methodology:** Kirstine Skov Benthien, Hanne Birke, Jon Trærup Andersen, Kristian Karstoft, Michaela Schiøtz, Charlotte Vermehren.

**Project administration:** Kirstine Skov Benthien, Laura Victoria Jedig Lech, Hanne Birke, Sidsel Maria Jørgensen, Tina Andersen, Stine Vest Hansen, Charlotte Vermehren.

**Resources:** Charlotte Vermehren.

**Supervision:** Kirstine Skov Benthien, Hanne Birke, Jon Trærup Andersen, Michaela Schiøtz, Charlotte Vermehren.

**Writing – original draft:** Kirstine Skov Benthien, Laura Victoria Jedig Lech, Hanne Birke, Sidsel Maria Jørgensen.

**Writing – review & editing:** Kirstine Skov Benthien, Laura Victoria Jedig Lech, Hanne Birke, Sidsel Maria Jørgensen, Tina Andersen, Stine Vest Hansen, Jon Trærup Andersen, Kristian Karstoft, Michaela Schiøtz, Charlotte Vermehren.

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
