## [Decision Letter · Decision Letter 0]

16 May 2023

PONE-D-22-27099Protocol for DEprescribing and Care to reduce Antipsychotics in Dementia (DECADE) – a hybrid effectiveness-implementation pilot studyPLOS ONE

Dear Dr. Benthien,

Thank you for submitting your manuscript to PLOS ONE. After careful consideration, we feel that it has merit but does not fully meet PLOS ONE’s publication criteria as it currently stands. Therefore, we invite you to submit a revised version of the manuscript that addresses the points raised during the review process.

We look forward to receiving your revised manuscript.

Kind regards,

Maw Pin Tan, M.D.

Academic Editor

PLOS ONE

Additional Editor Comments:

Apologies for taking so long, but it has been rather challenging securing reviewers.

Reviewers' comments:

Reviewer's Responses to Questions

**Comments to the Author**

1. Does the manuscript provide a valid rationale for the proposed study, with clearly identified and justified research questions?

Reviewer #1: Yes

Reviewer #2: Yes

2. Is the protocol technically sound and planned in a manner that will lead to a meaningful outcome and allow testing the stated hypotheses?

Reviewer #1: Yes

Reviewer #2: Partly

3. Is the methodology feasible and described in sufficient detail to allow the work to be replicable?

Reviewer #1: Yes

Reviewer #2: Yes

4. Have the authors described where all data underlying the findings will be made available when the study is complete?

Reviewer #1: No

Reviewer #2: No

5. Is the manuscript presented in an intelligible fashion and written in standard English?

Reviewer #1: Yes

Reviewer #2: Yes

6. Review Comments to the Author

You may also provide optional suggestions and comments to authors that they might find helpful in planning their study.

Reviewer #1: The authors describe a comprehensive proposal that is aimed at promoting the deprescribing of antipsychotic medications among nursing home residents with behavioural and psychological symptoms of dementia. The proposed research certainly has merit, but also raises a series of questions that warrant attention and clarification.

What concerns me most about this protocol is the following statement:

‘Since reduction of antipsychotics and use of non-pharmacological measures in persons with dementia are considered best practice the study does not need ethical board approval.’

I think this is a very far-ranging statement and I do not agree with it. In my experience, fast-track ethical approval may be granted for low-risk projects (e.g. involving review of de-identified, routinely-collected hospital data but no study procedures directly involving participants) and outright exemptions from ethical review for quality assurance projects (which should thus not be described as ‘prospective studies’). Even in the case of the latter, exemptions are communicated to researchers/authors in writing by the ethical review board. The assertion that studies that are in keeping with best practice guidelines do not require ethical board approval represents a slippery slope towards appropriate ethical oversight being routinely sidestepped in cases where this is far from justifiable. It will become apparent from the points I raise below that the present study falls into this category.

Somewhat confusingly, the authors then go on to state:

‘Due to cognitive impairment, the nursing home residents with dementia could not consent to the review of patient records. Therefore, the Centre for Regional Development has approved the review of patient records (R-22015601).’

The existence of a ‘Consent’ activity as part of pre-study screening in Figure 2 raises further questions about what this activity actually entails – i.e. whose consent is being obtained and for what purpose?

Fundamental to the issue of ethical oversight is whether this deprescribing initiative is one that the local authorities are planning to routinely and sustainably implement in their nursing homes (based on existing scientific evidence) in order to improve clinical practices, irrespective of whether it will be the focus of research in itself. If this is the case, the numerous assessments and activities that key personnel (especially medical and nursing staff) will be required to participate in may be justifiable as being a requirement of their jobs. If this is not the case, however, and this project is being primarily carried out as a research-focused ‘prospective study’ (as per the authors’ own description), questions are raised as to whether participation in the study’s procedures may be perceived as burdensome by treating team members and, in light of this, how their agreement or consent to be involved has been obtained. Are healthcare workers impacted by this project given a choice as to whether to participate or not? Why or why not? The issue of what family members might think about their loved ones’ antipsychotic medications being deprescribed is also not explicitly addressed in the study protocol. Will family member have any say in whether their relatives are enrolled in this study? Will family members who cannot be convinced about the benefits of deprescribing be allowed to exclude or withdraw their relatives from study participation if they believe that they are genuinely better off remaining on existing antipsychotic medications?

Even if the authors’ assertion that ethical board approval is not needed is true, I would expect the authors to obtain an exemption to this effect in writing from the applicable ethical board, given the vulnerable nature of the nursing home residents that are central to this project. Clarification as to how the decision-making body of the Centre for Regional Development differs from an ethical board, and why it is a suitable authority to be making research-related decisions, may also be helpful to readers.

In reality, the issue of deprescribing antipsychotic medications in nursing home residents with BPSD is very complex, despite the well-intended pressure applied by healthcare authorities on clinicians to do so. The possibility that hurried, inappropriate deprescribing might result in an exacerbation of BPSD, which, in extreme cases, might contribute to physical injuries among co-residents or staff, is an uncomfortable reality that may not be adequately acknowledged in the enthusiasm to deprescribe as a means of avoiding adverse drug effects. The assumption that there are always readily available and equally effective non-pharmacological alternatives for managing BPSD is an attractive one, but the fact that this may sometimes not hold true in more severe cases is another uncomfortable reality that is infrequently acknowledged. As both a clinical psychiatrist and hospital administrator working in this field, I am all too aware of the importance of balancing the competing priorities of the safety of residents on the one hand and that of co-residents and staff on the other. It is legitimate concerns such as these which may see medications being reinstated in some situations, as in the case of the HALT study (Aerts et al. 2019), and which potentially justify ethical board oversight. Hopefully, the authors of the present proposed study will not shy away from addressing such thorny issues during the implementation follow-up phase.

In the reviewer’s local practice, we have managed to attain deprescribing gains in nursing homes under our jurisdiction, but only after the implementation of a specialist multidisciplinary mental health team to provide intensive 1:1 support to facility nursing staff in caring for a select group of residents with more challenging BPSD. While GPs are the primary medication prescribers under this arrangement, psychotropic prescribing and deprescribing is ultimately driven (if not ‘directed’ in a strict sense) by the old age psychiatrist who is the clinical leader of this specialist team. Our experience suggests that in the absence of a specialist team to provide such intensive, direct support to facility staff (including GPs) on an ongoing basis, attempts to promote deprescribing and non-pharmacological treatment approaches may be resisted by facility nursing staff, especially where their prior training is predominantly medically, rather than psychiatrically, oriented.

From this perspective, I am a concerned that a 15-minute face-to-face education session (‘academic detailing’) for GPs might only service to increase their anxiety surrounding deprescribing, creating an urgency for them to do so while not fully equipping them with the skill set needed to authoritatively support nursing staff in implementing alternative, non-pharmacological approaches to managing BPSD. This is concerning given that it is GPs, rather than the geronto-psychiatrist, who will be making ‘the final decision regarding the changes to the medicine’. It is possible that the ‘municipal consultants with expertise in dementia’ alluded to by the authors might be able to fill in gaps in the knowledge and expertise of GPs to some degree. Nevertheless, it is important for doctors (as clinical leaders) who are engaged in deprescribing to have the skill set to effectively deal with the consequences of this action beyond simply resorting to re-prescribing the same or similar medications. On a positive note, the numerous reviews of medications and care plans that are planned at baseline and at the 3-, 6- and 9-month marks may serve to overcome the potentially limited benefits of only two education sessions to nursing staff, providing that all members of the treating team – including frontline nursing staff – are effectively engaged in this process (something which may be challenging due to competing workload priorities and staff availability due to shift work). Ideally such reviews should be conducted in a rigorous, educationally-oriented manner that is designed to promote the professional development of all the healthcare workers involved.

Despite the above reservations, I believe this project represents a potentially worthwhile contribution to our knowledge regarding antipsychotic medication deprescribing for BPSD in nursing home settings. I believe it is important for the authors to pursue and obtain either ethical review board approval or a formal written exemption in this regard – not only to safeguard study participants, but also their own prospects of completing an ethically acceptable, publishable study.

Refining the protocol in light of the above observations is also recommended, e.g.:

• If this project is considered to be a research study (rather than quality assurance project) by the ethical review board, consent should be sought from healthcare workers for their participation.

• GPs should receive more comprehensive training beyond the 15 minutes of ‘academic detailing’, not only in deprescribing but in both the pharmacological and non-pharmacological management of BPSD.

• Nurses would also benefit from ongoing nursing education to reinforce learning from the two sessions provided at the outset of the study. This could be integrated into the conduct of medication and care plan reviews.

• A clear position on how family members will be consulted in relation to their relatives participating in the study should be provided by the authors. If this project is deemed to be a research study by the ethical review board, the family members or legal guardians of the nursing home residents my need to consent to their participation.

Reviewer #2: Thank you for providing me with the opportunity to review this manuscript. It is a very interesting topic and highly relevant. I have some additional comments below

L55: This study will provide an indication of DECADE effectiveness. Is it a protocol or a pilot?

L74-77: This sentence is a bit long and it therefore becomes uncertain what is what. Try to split it up, so it becomes more clear what is the different parts of the non-pharmacological interventions.

L104-114: I'm missing some information on how long the study will run and maybe an argumentation as to why this type of study was chosen.

L119: is it both nursing homes and home nursing, and what is the difference?

L120-121: what is "staff skill mix"? Is it that there are staff with different educational levels or do they have different sets of skills concerning dementia care unrelated to their education or something else entirely?

L122-130: how was the participant selected? Who did the selection and when was it done?

L145: who is the medication consultant? Is it the same pharmacist that carries out the detailing?

L145-147: what guidelines and educational material is used? Is it developed by the research team or taken from someone else and tailored to fit this intervention?

L148-160: Education: who will do the education of the nursing home staff? Is the material used to educate developed by the research team? When will the education take place? When will the adapted session be carried out? Will this also be undertaken 4 times? How long will the sessions be?

L245-254: Assessment of BPDS: based on the text it is uncertain which NPI will be used. Consider rephrasing so that the NPI that is used in the study is mentioned firstly, and save the reasoning for not using the NPI-NH in the discussion.

L259-263: does the survey contain 56 items about demographics AND three components of the DeFHEAQ with each 16, 11 and 11 items? So how many items in total?

General:

I'm not an expert within the field of implementation science, and even though I have read the article by Bauer, I'm still a bit confused as to how an hybrid effectiveness-implementation pilot type two trial differentiate from a complex intervention with a process evaluation as described by the MRC framework by Skivington (Skivington K, Matthews L, Simpson SA, Craig P, Baird J, Blazeby JM, Boyd KA, Craig N, French DP, McIntosh E, Petticrew M, Rycroft-Malone J, White M, Moore L. A new framework for developing and evaluating complex interventions: update of Medical Research Council guidance. BMJ. 2021 Sep 30;374:n2061. doi: 10.1136/bmj.n2061. PMID: 34593508; PMCID: PMC8482308.).

When using MRC framework and defining this as a clinical, complex intervention, the trial may be registrered at Clinicaltrials.org and for structure for the protocol and ensuring substantial description of the intervention, SPIRIT and TiDiER checklists may be used.

Could you please elaborate a bit on the differences and why this study constitutes as the implementation study and what this means for the study?

To me it is a bit unclear how the data for the main outcome (50 % reduction in antipsychotic in 50 % of the population measued as DDD) is collected? Is this taken from the registries or do you have direct data from each patient from the medical charts?

Also in the "effectiveness and implementation" part (L288 and forward), it states that the effectiveness is assessed by BPSD. However is this only so, if there was a reduction of 50 % for a patient and there are an improvement or no-change in BPSD for that patient? Or how is the effectiveness of the drug reduction measured? And is it compared to baseline, the historical data or the follow-up times? What will you do if there is a reduction at month 3, but not at 9? What will count?

Also 50 % seems a lot - what if there isn't 50 % indication for reduction for the 60 patients included when going over the medical chart?

7. PLOS authors have the option to publish the peer review history of their article (what does this mean?). If published, this will include your full peer review and any attached files.

Reviewer #1: No

Reviewer #2: **Yes: **Pernille Hølmkjær

---

## [Author Response · Author response to Decision Letter 0]

27 Jun 2023

Response from the authors: The authors wish to thank the peer-reviewers for a thorough assessment of the manuscript. We have addressed each comment below and feel that the manuscript has improved during this process. 

Reviewers' comments:

Reviewer's Responses to Questions

Comments to the Author

1. Does the manuscript provide a valid rationale for the proposed study, with clearly identified and justified research questions?

Reviewer #1: Yes

Reviewer #2: Yes

2. Is the protocol technically sound and planned in a manner that will lead to a meaningful outcome and allow testing the stated hypotheses?

Reviewer #1: Yes

Reviewer #2: Partly

3. Is the methodology feasible and described in sufficient detail to allow the work to be replicable?

Reviewer #1: Yes

Reviewer #2: Yes

4. Have the authors described where all data underlying the findings will be made available when the study is complete?

Reviewer #1: No

Reviewer #2: No

Response from the authors: We have now specified in line 106 that since this manuscript describes a protocol without any results, there is no data to share in this manuscript.

5. Is the manuscript presented in an intelligible fashion and written in standard English?

Reviewer #1: Yes

Reviewer #2: Yes

6. Review Comments to the Author

You may also provide optional suggestions and comments to authors that they might find helpful in planning their study.

Reviewer #1: The authors describe a comprehensive proposal that is aimed at promoting the deprescribing of antipsychotic medications among nursing home residents with behavioural and psychological symptoms of dementia. The proposed research certainly has merit, but also raises a series of questions that warrant attention and clarification.

What concerns me most about this protocol is the following statement:

‘Since reduction of antipsychotics and use of non-pharmacological measures in persons with dementia are considered best practice the study does not need ethical board approval.’

I think this is a very far-ranging statement and I do not agree with it. In my experience, fast-track ethical approval may be granted for low-risk projects (e.g., involving review of de-identified, routinely-collected hospital data but no study procedures directly involving participants) and outright exemptions from ethical review for quality assurance projects (which should thus not be described as ‘prospective studies’). Even in the case of the latter, exemptions are communicated to researchers/authors in writing by the ethical review board. The assertion that studies that are in keeping with best practice guidelines do not require ethical board approval represents a slippery slope towards appropriate ethical oversight being routinely sidestepped in cases where this is far from justifiable. It will become apparent from the points I raise below that the present study falls into this category.

Response from the authors: Thank you for allowing us to clarify this statement. After submission of the manuscript, we received documentation from the board of ethics that this study does indeed not require ethical committee approval according to Danish legislation – detailed in the Ethics section. We agree with the reviewer, that these types of studies would absolutely benefit from ethical approval, but Danish legislation will have it otherwise. 

Somewhat confusingly, the authors then go on to state:

‘Due to cognitive impairment, the nursing home residents with dementia could not consent to the review of patient records. Therefore, the Centre for Regional Development has approved the review of patient records (R-22015601).’

The existence of a ‘Consent’ activity as part of pre-study screening in Figure 2 raises further questions about what this activity actually entails – i.e. whose consent is being obtained and for what purpose?

Fundamental to the issue of ethical oversight is whether this deprescribing initiative is one that the local authorities are planning to routinely and sustainably implement in their nursing homes (based on existing scientific evidence) in order to improve clinical practices, irrespective of whether it will be the focus of research in itself. If this is the case, the numerous assessments and activities that key personnel (especially medical and nursing staff) will be required to participate in may be justifiable as being a requirement of their jobs. If this is not the case, however, and this project is being primarily carried out as a research-focused ‘prospective study’ (as per the authors’ own description), questions are raised as to whether participation in the study’s procedures may be perceived as burdensome by treating team members and, in light of this, how their agreement or consent to be involved has been obtained. Are healthcare workers impacted by this project given a choice as to whether to participate or not? Why or why not? 

Response from the authors: We have now clarified in the Ethics sections, that Ethical board approval or exemptions thereof covers the intervention and data handling also require consent or administrative approval. Since this is an implementation study, the intervention activities are part of healthcare professionals’ tasks.

The issue of what family members might think about their loved ones’ antipsychotic medications being deprescribed is also not explicitly addressed in the study protocol. Will family member have any say in whether their relatives are enrolled in this study? Will family members who cannot be convinced about the benefits of deprescribing be allowed to exclude or withdraw their relatives from study participation if they believe that they are genuinely better off remaining on existing antipsychotic medications?

Response from the authors: Thank you for pointing to the importance of loved ones. We have discussed the role of informal caregivers at length in the group under the ideal that informal caregivers should always be included in patients’ care and the realization that few are willing and able to. The decision to offer involvement has now been specified under Medication Review in network groups, lines 180-186, and Care plans, lines 188-189, respectively. 

Even if the authors’ assertion that ethical board approval is not needed is true, I would expect the authors to obtain an exemption to this effect in writing from the applicable ethical board, given the vulnerable nature of the nursing home residents that are central to this project. Clarification as to how the decision-making body of the Centre for Regional Development differs from an ethical board, and why it is a suitable authority to be making research-related decisions, may also be helpful to readers.

Response from the authors: We have now clarified in the Ethics sections, that Ethical board approval or exemptions thereof covers the intervention and data handling also require consent or administrative approval by the Centre for Regional Development.

In reality, the issue of deprescribing antipsychotic medications in nursing home residents with BPSD is very complex, despite the well-intended pressure applied by healthcare authorities on clinicians to do so. The possibility that hurried, inappropriate deprescribing might result in an exacerbation of BPSD, which, in extreme cases, might contribute to physical injuries among co-residents or staff, is an uncomfortable reality that may not be adequately acknowledged in the enthusiasm to deprescribe as a means of avoiding adverse drug effects. The assumption that there are always readily available and equally effective non-pharmacological alternatives for managing BPSD is an attractive one, but the fact that this may sometimes not hold true in more severe cases is another uncomfortable reality that is infrequently acknowledged. As both a clinical psychiatrist and hospital administrator working in this field, I am all too aware of the importance of balancing the competing priorities of the safety of residents on the one hand and that of co-residents and staff on the other. It is legitimate concerns such as these which may see medications being reinstated in some situations, as in the case of the HALT study (Aerts et al. 2019), and which potentially justify ethical board oversight. Hopefully, the authors of the present proposed study will not shy away from addressing such thorny issues during the implementation follow-up phase.

Response from the authors: Thank you for sharing this excellent insight and experience. We have clarified our choice of primary outcome under Design and Methods, line, 109-111 with the following statement: The choice of a 50% reduction in 50% of nursing home residents instead of 100% / 100% reflects the realization that, in some cases with severe BPSD or pre-existing psychiatric disorders, deprescription is not in the best interest of the nursing home resident. 

In the reviewer’s local practice, we have managed to attain deprescribing gains in nursing homes under our jurisdiction, but only after the implementation of a specialist multidisciplinary mental health team to provide intensive 1:1 support to facility nursing staff in caring for a select group of residents with more challenging BPSD. While GPs are the primary medication prescribers under this arrangement, psychotropic prescribing and deprescribing is ultimately driven (if not ‘directed’ in a strict sense) by the old age psychiatrist who is the clinical leader of this specialist team. Our experience suggests that in the absence of a specialist team to provide such intensive, direct support to facility staff (including GPs) on an ongoing basis, attempts to promote deprescribing and non-pharmacological treatment approaches may be resisted by facility nursing staff, especially where their prior training is predominantly medically, rather than psychiatrically, oriented.

Response from the authors: We share some of those experiences and are also concerned about potential resistance from nursing staff. We have specified in line 160-162 about the education sessions, that a geronto-psychiatrist is part of the multidisciplinary team that educates facility staff and in line 177-179 about the conduct of medication review for each resident as an educational-oriented intervention, respectively.

From this perspective, I am a concerned that a 15-minute face-to-face education session (‘academic detailing’) for GPs might only service to increase their anxiety surrounding deprescribing, creating an urgency for them to do so while not fully equipping them with the skill set needed to authoritatively support nursing staff in implementing alternative, non-pharmacological approaches to managing BPSD. This is concerning given that it is GPs, rather than the geronto-psychiatrist, who will be making ‘the final decision regarding the changes to the medicine’. It is possible that the ‘municipal consultants with expertise in dementia’ alluded to by the authors might be able to fill in gaps in the knowledge and expertise of GPs to some degree. Nevertheless, it is important for doctors (as clinical leaders) who are engaged in deprescribing to have the skill set to effectively deal with the consequences of this action beyond simply resorting to re-prescribing the same or similar medications. On a positive note, the numerous reviews of medications and care plans that are planned at baseline and at the 3-, 6- and 9-month marks may serve to overcome the potentially limited benefits of only two education sessions to nursing staff, providing that all members of the treating team – including frontline nursing staff – are effectively engaged in this process (something which may be challenging due to competing workload priorities and staff availability due to shift work). Ideally such reviews should be conducted in a rigorous, educationally-oriented manner that is designed to promote the professional development of all the healthcare workers involved.

Response from the authors: We share the concern about the possibility of re-prescribing or omission of enacting the decisions in medication review and hope that the comprehensive and multidisciplinary intervention including medication review in network groups will suffice in overcoming the obstacles of successfully deprescribing antipsychotics. Hence, the deprescribing education consists of both the 15-minute academic detailing and subsequently the conduction of medication reviews in an educational-oriented manner. Since prescription of antipsychotics is not formally reserved to psychiatry, the role of the GPs is decided in agreement between the GPs union and regions and not subject to change by these authors. We have specified this in line 179. 

Despite the above reservations, I believe this project represents a potentially worthwhile contribution to our knowledge regarding antipsychotic medication deprescribing for BPSD in nursing home settings. I believe it is important for the authors to pursue and obtain either ethical review board approval or a formal written exemption in this regard – not only to safeguard study participants, but also their own prospects of completing an ethically acceptable, publishable study.

Response from the authors: Thank you, we have obtained the documentation and specified this in the Ethics section.

Refining the protocol in light of the above observations is also recommended, e.g.:

• If this project is considered to be a research study (rather than quality assurance project) by the ethical review board, consent should be sought from healthcare workers for their participation.

Response from the authors: We have consent from healthcare workers for data handling and specified this in the Ethics section.

• GPs should receive more comprehensive training beyond the 15 minutes of ‘academic detailing’, not only in deprescribing but in both the pharmacological and non-pharmacological management of BPSD.

Response from the authors: We would like to, but in Denmark the allocated education time for a GP is scarce due to financial agreements. In this intervention, however, the GPs receive education regarding deprescribing, pharmacological and non-pharmacological treatment with antipsychotics during the 15-minute academic detailing intervention as well as during the educational-oriented medication review network meetings.

• Nurses would also benefit from ongoing nursing education to reinforce learning from the two sessions provided at the outset of the study. This could be integrated into the conduct of medication and care plan reviews.

Response from the authors: Yes, we agree that the two educational sessions are far from sufficient. We have emphasized the educational value in the process of conducting medication review and creating the care plans, see line 175-176 and 194-195.

• A clear position on how family members will be consulted in relation to their relatives participating in the study should be provided by the authors. If this project is deemed to be a research study by the ethical review board, the family members or legal guardians of the nursing home residents my need to consent to their participation.

Response from the authors: We have added this to line 188 and elaborated on the ethical issues in lines 317-331. 

Reviewer #2: Thank you for providing me with the opportunity to review this manuscript. It is a very interesting topic and highly relevant. I have some additional comments below

L55: This study will provide an indication of DECADE effectiveness. Is it a protocol or a pilot?

Response from the authors: We have clarified that this article is a protocol for a pilot.

L74-77: This sentence is a bit long and it therefore becomes uncertain what is what. Try to split it up, so it becomes more clear what is the different parts of the non-pharmacological interventions.

Response from the authors: Thank you for noticing, we have split it up.

L104-114: I'm missing some information on how long the study will run and maybe an argumentation as to why this type of study was chosen.

Response from the authors: We have now added study length in line 101 and argumentation in line 115-117 in Design and outcomes.

L119: is it both nursing homes and home nursing, and what is the difference?

Response from the authors: We have now specified that only nursing homes are included in the study (home nursing is in the general description of primary care).

L120-121: what is "staff skill mix"? Is it that there are staff with different educational levels or do they have different sets of skills concerning dementia care unrelated to their education or something else entirely?

Response from the authors: We have now added that skill mix covers the mix of registered nurses, nurses’ aides, and allied health professionals.

L122-130: how was the participant selected? Who did the selection and when was it done?

Response from the authors: We have now specified in line 132 that all residents in all nursing homes in Hillerød are included.

L145: who is the medication consultant? Is it the same pharmacist that carries out the detailing?

Response from the authors: Yes, we have now changed the wording to pharmacist to avoid confusion.

L145-147: what guidelines and educational material is used? Is it developed by the research team or taken from someone else and tailored to fit this intervention?

Response from the authors: We have now specified, that the material is developed for this project.

L148-160: Education: who will do the education of the nursing home staff? Is the material used to educate developed by the research team? When will the education take place? When will the adapted session be carried out? Will this also be undertaken 4 times? How long will the sessions be?

Response from the authors: We have now included those details under Education, lines 164-167, except for the adapted session that has not been decided at the time of writing. 

L245-254: Assessment of BPDS: based on the text it is uncertain which NPI will be used. Consider rephrasing so that the NPI that is used in the study is mentioned firstly, and save the reasoning for not using the NPI-NH in the discussion.

Response from the authors: Yes, we can see how it is unclear since we mention several NPI versions. We have now deleted the 

L259-263: does the survey contain 56 items about demographics AND three components of the DeFHEAQ with each 16, 11 and 11 items? So how many items in total?

Response from the authors: We have now specified that the 56 items are the total, thank you for noticing this.

General:

I'm not an expert within the field of implementation science, and even though I have read the article by Bauer, I'm still a bit confused as to how an hybrid effectiveness-implementation pilot type two trial differentiate from a complex intervention with a process evaluation as described by the MRC framework by Skivington (Skivington K, Matthews L, Simpson SA, Craig P, Baird J, Blazeby JM, Boyd KA, Craig N, French DP, McIntosh E, Petticrew M, Rycroft-Malone J, White M, Moore L. A new framework for developing and evaluating complex interventions: update of Medical Research Council guidance. BMJ. 2021 Sep 30;374:n2061. doi: 10.1136/bmj.n2061. PMID: 34593508; PMCID: PMC8482308.).

When using MRC framework and defining this as a clinical, complex intervention, the trial may be registrered at Clinicaltrials.org and for structure for the protocol and ensuring substantial description of the intervention, SPIRIT and TiDiER checklists may be used.

Could you please elaborate a bit on the differences and why this study constitutes as the implementation study and what this means for the study?

Response from the authors: We agree, this is a complex intervention. The MRC council works detail four research phases, the last of which is implementation. Since the study is not a trial designed with a control group, the trial registry and guidelines are not applicable. 

To me it is a bit unclear how the data for the main outcome (50 % reduction in antipsychotic in 50 % of the population measued as DDD) is collected? Is this taken from the registries or do you have direct data from each patient from the medical charts?

Response from the authors: We have now specified that use of antipsychotic drugs are available from medical charts.

Also in the "effectiveness and implementation" part (L288 and forward), it states that the effectiveness is assessed by BPSD. However is this only so, if there was a reduction of 50 % for a patient and there are an improvement or no-change in BPSD for that patient? Or how is the effectiveness of the drug reduction measured? And is it compared to baseline, the historical data or the follow-up times? What will you do if there is a reduction at month 3, but not at 9? What will count?

Also 50 % seems a lot - what if there isn't 50 % indication for reduction for the 60 patients included when going over the medical chart?

Response from the authors: Thank you for raising the important implications of multiple outcomes in complex interventions. We have now elaborated on the reasoning behind the choice of primary outcome under Design and outcomes and have elaborated on the simultaneous focus on antipsychotics reduction while maintaining or improving BPSD, lines 111-114.

---

## [Decision Letter · Decision Letter 1]

25 Oct 2023

Protocol for DEprescribing and Care to reduce Antipsychotics in Dementia (DECADE) – a hybrid effectiveness-implementation pilot study

PONE-D-22-27099R1

Dear Dr. Benthien,

We’re pleased to inform you that your manuscript has been judged scientifically suitable for publication and will be formally accepted for publication once it meets all outstanding technical requirements.

Kind regards,

Maw Pin Tan, M.D.

Academic Editor

PLOS ONE

Additional Editor Comments (optional):

Reviewers' comments:

Reviewer's Responses to Questions

**Comments to the Author**

1. Does the manuscript provide a valid rationale for the proposed study, with clearly identified and justified research questions?

Reviewer #1: Yes

Reviewer #2: Yes

2. Is the protocol technically sound and planned in a manner that will lead to a meaningful outcome and allow testing the stated hypotheses?

Reviewer #1: Yes

Reviewer #2: Yes

3. Is the methodology feasible and described in sufficient detail to allow the work to be replicable?

Reviewer #1: Yes

Reviewer #2: Yes

4. Have the authors described where all data underlying the findings will be made available when the study is complete?

Reviewer #1: Yes

Reviewer #2: Yes

5. Is the manuscript presented in an intelligible fashion and written in standard English?

Reviewer #1: Yes

Reviewer #2: Yes

6. Review Comments to the Author

You may also provide optional suggestions and comments to authors that they might find helpful in planning their study.

Reviewer #1: Thank you to the authors for carefully considering my feedback (Review #1) and revising the manuscript accordingly. I am satisfied with the authors’ responses and the changes that have been made to the manuscript, which have improved its quality.

I note the authors’ response that ‘in Denmark the allocated education time for a GP is scarce due to financial agreements.’ I suggest that it would be helpful to include a statement to this effect in the manuscript itself in the section on ‘Academic Detailing’, by way of emphasising to readers that the researchers are doing everything possible to improve the knowledge and skills of GPs, within the limitations of system constraints that are beyond their control. In this context, the Academic Detailing intervention may be seen as all the more important and powerful.

Based on the authors’ responses to my feedback, I am happy for this manuscript to proceed to publication. I will leave it to Reviewer #2 to comment on whether their feedback has been adequately addressed by the authors.

I hope the authors are successful in carrying out this study and publishing their findings and extend my best wishes to them in this regard.

Reviewer #2: The authors have responded to the raised issues and changed accordingly in the manuscript in a satisfactory way.

7. PLOS authors have the option to publish the peer review history of their article (what does this mean?). If published, this will include your full peer review and any attached files.

Reviewer #1: No

Reviewer #2: No

---

## [Editor Report · Acceptance letter]

31 Oct 2023

PONE-D-22-27099R1 

Protocol for DEprescribing and Care to reduce Antipsychotics in Dementia (DECADE) – a hybrid effectiveness-implementation pilot study 

Dear Dr. Benthien:

I'm pleased to inform you that your manuscript has been deemed suitable for publication in PLOS ONE. Congratulations! Your manuscript is now with our production department. 

Kind regards, 

on behalf of

Dr. Maw Pin Tan 

Academic Editor

PLOS ONE